# Dispersal and fire limit Arctic shrub expansion

Yanlan Liu [1,2 ✉], William J. Riley [3], Trevor F. Keenan [3,4], Zelalem A. Mekonnen [3], Jennifer A. Holm[3], Qing Zhu [3] & Margaret S. Torn [3]

Arctic shrub expansion alters carbon budgets, albedo, and warming rates in high latitudes but remains challenging to predict due to unclear underlying controls. Observational studies and models typically use relationships between observed shrub presence and current environmental suitability (bioclimate and topography) to predict shrub expansion, while omitting shrub demographic processes and non-stationary response to changing climate. Here, we use high-resolution satellite imagery across Alaska and western Canada to show that observed shrub expansion has not been controlled by environmental suitability during 1984–2014, but can only be explained by considering seed dispersal and fire. These findings provide the impetus for better observations of recruitment and for incorporating currently under-represented processes of seed dispersal and fire in land models to project shrub expansion and climate feedbacks. Integrating these dynamic processes with projected fire extent and climate, we estimate shrubs will expand into 25% of the non-shrub tundra by 2100, in contrast to 39% predicted based on increasing environmental suitability alone. Thus, using environmental suitability alone likely overestimates and misrepresents shrub expansion pattern and its associated carbon sink.

[1] School of Earth Sciences, The Ohio State University, Columbus, OH, USA. [2] School of Environment and Natural Resources, The Ohio State University, Columbus, OH, USA. [3] Climate and Ecosystem Sciences Division, Lawrence Berkeley National Laboratory, Berkeley, CA, USA. [4] Department of Environmental Science Policy and Management, University of California, Berkeley, CA, USA. ✉email: liu.9367@osu.edu

The Arctic has warmed more than twice as fast as the global average and is projected to continue outpacing lower latitudes over the 21st century[1]. Rapid climate warming in recent decades and associated feedbacks have led to shifts in Arctic vegetation composition and abundance[2–4]. In particular, increased tundra shrub cover has been widely observed through field surveys[5], aerial photographs[6,7], and satellite remote sensing[8,9]. Pervasive shrub expansion can heat the atmosphere through decreased albedo and increased greenhouse warming induced by atmospheric water vapor, resulting from increased evapotranspiration and regional ocean feedbacks[10–12]. Locally, shrubs can warm the soil in the winter due to the insulation of accumulated snow, which deepens the active layer and accelerates soil carbon loss compared to non-shrub tundra[13,14]. Moreover, the distribution of shrubs also affects nutrient cycling, animal populations[15], and wildfire risk and associated carbon emissions[16,17]. Understanding controls of shrub expansion patterns is therefore crucial to predicting climate feedbacks and ecological consequences of the rapidly changing Arctic.

The area where temperature limits the growth of Arctic vegetation has been declining over the past decades[18]. Increasing temperature has been identified as a major control of shrub expansion[5,19,20]. However, the influence of temperature can be attenuated or reversed by soil moisture limitation, snow distribution, and topography[4,21–25]. The majority of observational-based studies focus on environmental factors and attribute the heterogeneity of shrub expansion to spatial variation of environment-based suitability, i.e., the likelihood of shrub presence given environmental conditions[26]. Based on space-for-time substitutions, some of those studies used derived spatial environment-vegetation relationships to assess future shrub expansion[5,20,21], assuming stationary relationships between species and the environment. Although this approach has been found effective in predicting species distributions when ecosystems are in dynamic equilibrium, e.g., under a relatively stable climate or over a sufficiently long time scale[27], it ignores transient responses and non-stationary ecological processes, thus causing errors in projected ecosystem change[28–30]. As the Arctic tundra deviates from the historical quasi-equilibrium due to climate change, evaluating the dynamic roles of plant migration and disturbance becomes especially relevant.

With changes in growing conditions under a warmer climate, the successful establishment of new shrub patches depends on seed dispersal. Seeds can be dispersed through many biotic and abiotic vectors, such as gravity, animals, wind, ocean currents, and drifting sea ice, which result in dispersal ranges from meters to hundreds of kilometers[31,32]. Seed dispersal has been investigated in previous studies to estimate species range shifts[32–34], and was found important in explaining shifts in vegetation composition at sites in alpine[35], mediterranean[36,37], and tropical biomes[38,39]. Nonetheless, the impact of seed dispersal on vegetation patterns is also compounded by suitable environmental niches and thus is not always the limiting factor[40–42]. In the Arctic, long-distance seed dispersal is a critical mechanism affecting species distribution. Genetic analysis has revealed repeated long-distance seed dispersal to a remote archipelago from multiple source regions since the last glacial retreat, while the resulting species distribution is predominantly shaped by temperature that limits environmental suitability for establishment[41]. In contrast to a relatively stable climate over the past several millennia, the fast-changing climate over recent decades might lead to shifts in the relative dominance of environmental suitability and seed dispersal in shaping Arctic shrub expansion.

The dynamic trajectory of ecosystems may also be affected by disturbance. Although historically rare in Arctic ecosystems, wildfire is expected to become more intense and frequent as the climate warms[16,43]. In the short term, wildfires may cause seed and seedling mortality, which could limit post-fire recruitment. On the other hand, wildfires can alter post-fire vegetation trajectories by heating the soil during the fire, cause long-term soil warming by removing surface litter, and improve seedbed nutrient availability, thus facilitating germination and seedling establishment[44–48]. For example, modeling and site-based field studies have reported both enhanced expansion and diminished recovery of shrubs from four years to two decades after wildfires at several sites in Alaska[45,47,49–51]. How wildfires affect shrub expansion across large gradients of environmental suitability and seed dispersal has barely been evaluated using observations.

We focused on shrub expansion from 1984 to 2014 across the northwestern region of North America covering Alaska and western Canada, i.e., the NASA Arctic-Boreal Vulnerability Experiment (ABoVE) core domain. Shrub expansion was detected using an annual dominant land-cover product derived from Landsat surface reflectance and trained over field photography and very high-resolution imagery[3]. Areas classified as shrubland include prostrate dwarf-shrub tundra and erect-shrub tundra[52], dominated by species of birch (Betula spp.), alder (Alnus spp.), willow (Salix spp.), and other dwarf evergreen and semi-deciduous shrubs. Field surveys have detected expansion of these shrub communities[53–55]. Here, shrub expansion is defined as shrub dominance in tundra originally dominated by non-woody species at a 30 m scale. We collected topographic and regionally downscaled bioclimatic variables across the domain to identify the variables most informative for observed shrub expansion ("Methods"). Based on these selected topographic and bioclimatic conditions, averaged over three decades prior to 1984, we estimated environmental suitability for shrubs in 1984 using a random forest model. The same model was then used to calculate the environmental suitability for shrubs in 2014 using 1985–2014 average bioclimatic conditions. We analyzed whether changes in environmental suitability could explain shrub expansion from 1984 to 2014. Seed-arrival probability, a measure of spatial proximity to existing shrub patches, was calculated through convolution of seed-dispersal kernels over 1984 shrub cover images. Given the variety of dispersal mechanisms, we considered both short- and long-distance dispersal kernels, and optimized the range and shape parameters to fit observed shrub expansion. The year and location of fires were obtained from a Landsat-derived annual burn scar product. We investigated individual and compound impacts of environmental suitability, seed dispersal, and fire occurrence on observed 1984–2014 shrub expansion. Based on the resulting sensitivities and the projection of bioclimatic conditions and fire from climate models, we estimated shrub expansion in 2040, 2070, and 2100, and explored the relative importance of environmental suitability change, fire, and seed dispersal on projected shrub expansion. We found observed shrub expansion did not follow the pattern of environmental suitability but can only be explained by considering seed dispersal and fire. Shrub expansion under the projected climate is likely overestimated if neglecting the limitation of seed dispersal and fire.

## Results

**Environmental suitability of shrubs**. We first estimated environmental suitability for shrubs as of 1984. Among the 27 variables, the seven most informative variables, as identified based on variance inflation factors (Supplementary Table 1 and Supplementary Fig. 1), were three bioclimatic variables (degree-days above 5 °C, annual heat-moisture index, and precipitation as snow) and four topographic variables (elevation, slope, aspect,

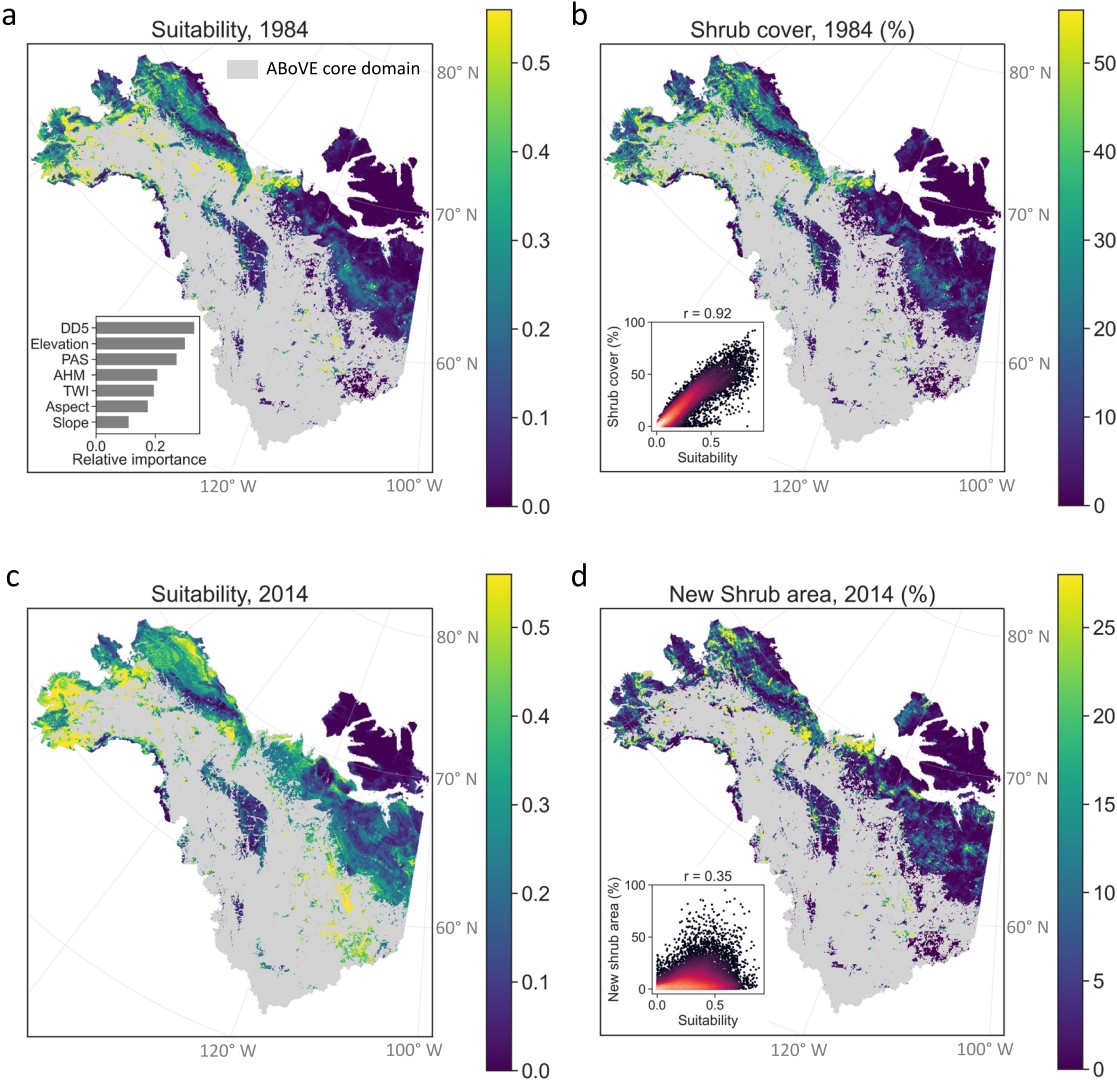

**Fig. 1 Environmental suitability did not explain shrub expansion between 1984 and 2014.** Environmental suitability of shrubs in (**a**) 1984 and (**c**) 2014 estimated using topographic and bioclimatic conditions, including annual degree-days above 5 °C (DD5), annual degree precipitation as snow (PAS), annual heat-moisture index (AHM), elevation, slope, aspect, and topographic wetness index (TWI). The inset of (**a**) shows the relative importance of these factors on environmental suitability. **b** Fraction of shrub cover in 1984. **d** Fraction of new shrub area at a 4 km scale in 2014, i.e., non-shrub tundra in 1984 that became dominated by shrubs by 2014. The insets of (**b**, **d**) show the corresponding relationships with environmental suitability, where brighter colors represent higher dot density. The gray areas are dominated by land-cover types other than shrubs and non-woody plants and are excluded from the analyses.

and topographic wetness index). Based on the random forest model, degree-days above 5 °C were the most important variable for environmental suitability in 1984, followed by elevation and precipitation as snow (Fig. 1a). In terms of the direction and shape of response, higher environmental suitability was associated with higher degree-days above 5 °C, lower elevation, and higher precipitation as snow (Supplementary Fig. 2), although environmental suitability responds to these variables non-monotonically across different combinations of climate and topographic conditions (Supplementary Fig. 3). The non-monotonic responses could be partially attributable to the coexistence of multiple shrub species that have different optimal environmental conditions, and regional collinearity among bioclimatic and topographic conditions that may not be completely disentangled using a data-driven approach.

Environmental suitability in 1984 was higher in southwestern Alaska, eastern Seward Peninsula, and northern Northwest Territories of Canada, but lower on the northern edge of the

North Slope of Alaska and northern Canada and mountainous regions such as the Brooks Range and the Mackenzie Mountains (Fig. 1a, reference locations noted in Supplementary Fig. 1c). This pattern of environmental suitability was largely consistent with observed shrub distribution in 1984 (Pearson's $r = 0.92$, Fig. 1b), suggesting that environmental suitability alone explains shrub distribution under quasi-equilibrium conditions. Due to climate warming since 1984, the domain became more suitable in 2014 on average, especially in southwestern Alaska, eastern Seward Peninsula, the North Slope, and the southeast of the domain (Fig. 1c). The area with high environmental suitability (>0.4) increased from 13.4% to 28.3% of the region. However, these highly suitable regions experienced limited shrub expansion (Fig. 1d). Instead, hot spots of shrub expansion were found in the west of the North Slope and northwestern Canada. Across the entire domain, environmental suitability was much less related to new (i.e., expanded) shrub area in 2014 ($r = 0.35$) than to existing shrub cover in 1984 ($r = 0.92$). Accounting for different initial

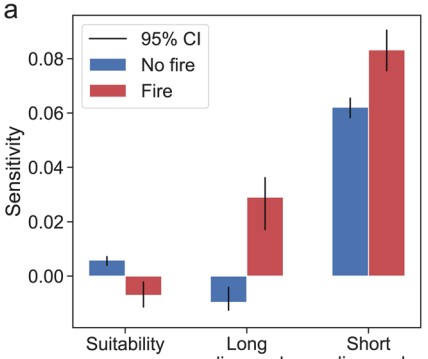
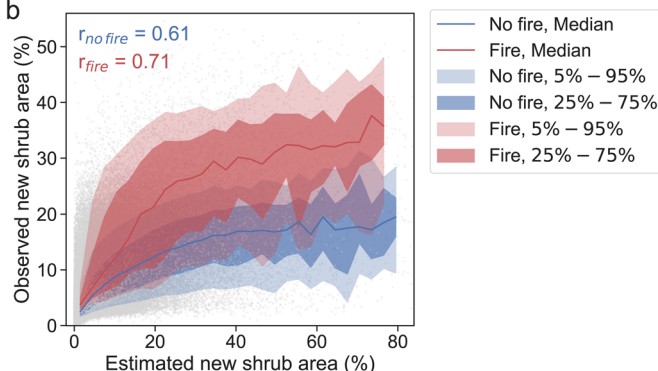

**Fig. 2 Dispersal and fire explain shrub expansion during 1984–2014. a** Sensitivity of new shrub area to environmental suitability and probabilities of seed arrival via short- and long-distance dispersal at locations with (red bars) and without (blue bars) fire. Colored bars denote the average and vertical black lines denote range of the 95% confidence interval of the regression coefficients across optimal dispersal kernel parameters ($n = 533$). **b** Observed and estimated new shrub area, i.e., areal fraction at a 4 km scale, in 2014 at locations with (red) and without (blue) fire. The lines and shaded bands represent the medians and ranges of observed new shrub area (gray dots) for each bin of estimates with a width of 2%.

land-cover types of the non-woody tundra and change in environmental suitability also barely contributed to explaining the pattern of shrub expansion (Supplementary Fig. 4). These results suggest environmental suitability was not the major limiting factor of shrub expansion between 1984 and 2014.

**Impacts of seed dispersal and fire**. The best-fitting long-distance dispersal was represented using a fat-tail kernel ($c = 0.5$ in Eq. (1) in "Methods") with a range parameter of 39 km (Supplementary Fig. 5); and the short-distance dispersal was best represented using an exponential power kernel ($c = 1.5$ in Eq. (1) in "Methods"), which is between an exponential kernel and a Gaussian kernel and has a range parameter of 600 m. Notably, in regions not disturbed by fire, the area fraction of shrub expansion was the most sensitive to short-distance dispersal, 9.5 times more sensitive than to environmental suitability based on the regression coefficients (Fig. 2a). The weak negative sensitivity to long-distance dispersal likely arose from the trade-off between the sensitivities to short- and long-distance dispersal, which might not be precisely separated based on the data due to their spatial correlation ($r = 0.71$). However, both long- and short-distance dispersal became significantly more important in facilitating shrub expansion after fire, compared to areas without fire (Fig. 2a). The median of the sensitivities to long- and short-distance dispersal increased from −0.010 to 0.029 and from 0.062 to 0.083, respectively, for areas that experienced fire; whereas the sensitivity to environmental suitability reduced from 0.006 to −0.007. Across the entire domain, fire disturbance enhanced the likelihood of shrub expansion, especially in highly suitable areas with high seed-arrival probability (Fig. 2b). Accounting for seed-arrival probability and environmental suitability improved the estimation accuracy of shrub expansion from r = 0.35 (Fig. 1d) to $r = 0.61$ (areas with fire, 7.8 km²) and r = 0.71 (areas without fire, 149.1 km²) (Fig. 2b). We note that the shrub expansion pattern can also be influenced by other factors unaccounted for, leading to a spatial correlation pattern unexplained by the considered covariates[56]. Nonetheless, additionally accounting for spatial correlation of shrub expansion patterns using a spatial regression (Eq. (4) in "Methods") only slightly improved estimation accuracy but did not fundamentally alter the estimated sensitivities (Supplementary Fig. 6). These findings show that, over recent decades, dispersal has been a stronger limiting factor than environmental suitability on shrub expansion. Non-woody tundra locations becoming more suitable is not sufficient for shrub expansion to occur. By contrast, fire disturbance and proximity to

existing shrub patches make shrub expansion more likely in the newly suitable areas.

**Predicted shrub expansion**. Across the domain, 6.8% of non-shrub tundra in 1984 had become dominated by shrubs by 2014. Using our established relationships (Fig. 2), we estimated that the shrubified area fraction would increase progressively to 25.1% ± 3.0% by 2100 (Fig. 3a and Supplementary Fig. 7) corresponding to 253,651 ± 30,317 km² more shrub cover than in 2014, with the uncertainty originating from uncertainty in the empirically derived sensitivities (Fig. 2a). The results suggest substantial shrub expansion in southwestern Alaska, southern and eastern Seward Peninsula, south and north of the Brooks Range, and northern Northwest Territories of Canada. The Victoria Island, western Nunavut, the Brooks Range, and the Mackenzie Mountains will likely experience limited shrub expansion. Note that the projected shrub expansion estimated here originates from the combined impacts of environmental suitability under projected climate change, seed dispersal, and projected burn area. The resulting pattern (Fig. 3a) does not account for shrub loss due to competition, pests, and herbivores[3,25,57,58], which are beyond the scope of our study. By contrast, without considering the impact of dispersal and fire, the relationship between shrub presence and increased environmental suitability alone (Fig. 1a, b) predicts a higher fraction (38.9%) of non-shrub tundra in 1984 will become shrublands by 2100. Notably, the shrub expansion pattern predicted using environmental suitability alone shows substantial increase of shrub cover in the North Slope and northern Canada (Fig. 3b), which is significantly different from the expansion if dispersal and fire limitations are considered (Fig. 3a). Thus, relying on environmental suitability alone likely results in predictions that overestimate shrub expansion and misrepresent the spatial patterns. As a result, observational studies and models that project shrub expansion without considering the biological and physical constraints of dispersal and fire likely overestimate the 21st-century carbon sink in the Arctic tundra due to shrub responses to warming.

**Relative impact of environmental suitability, fire, and dispersal**. We investigated the spatial patterns of projected changes in environmental suitability, burn area, and seed-arrival probability in 2100, and used synthetic scenarios, i.e., turning off one factor at a time, to disentangle their individual impacts on projected shrub expansion shown in Fig. 3 (see "Methods"). Compared to 1984, environmental suitability in 2100 increased in

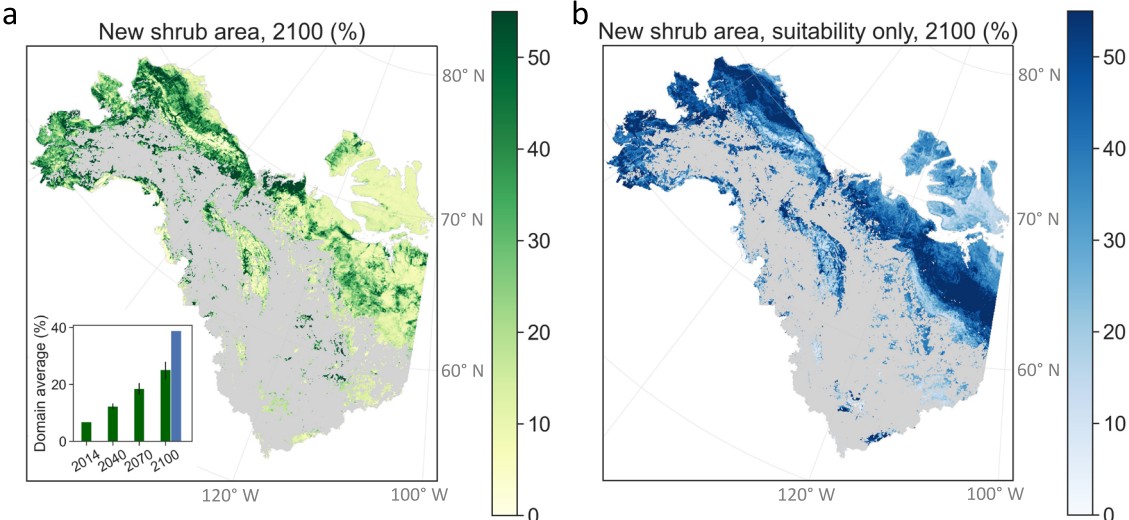

**Fig. 3 A quarter of non-woody tundra in 1984 will be colonized by shrubs by 2100 based on the climate scenario RCP8.5.** Spatial pattern of new shrub area in 2100 predicted using (**a**) environmental suitability, dispersal, and fire, and (**b**) environmental suitability alone. The inset of (**a**) shows the domain average of new shrub area predicted with (green bars) and without (blue bar) considering dispersal and fire. The vertical black lines span the upper and lower boundaries due to the uncertainty of the estimated sensitivities ($n = 533$, see "Methods" for details), i.e., vertical black lines in Fig. 2a. Using environmental suitability alone overestimates shrub expansion and misrepresents the spatial pattern.

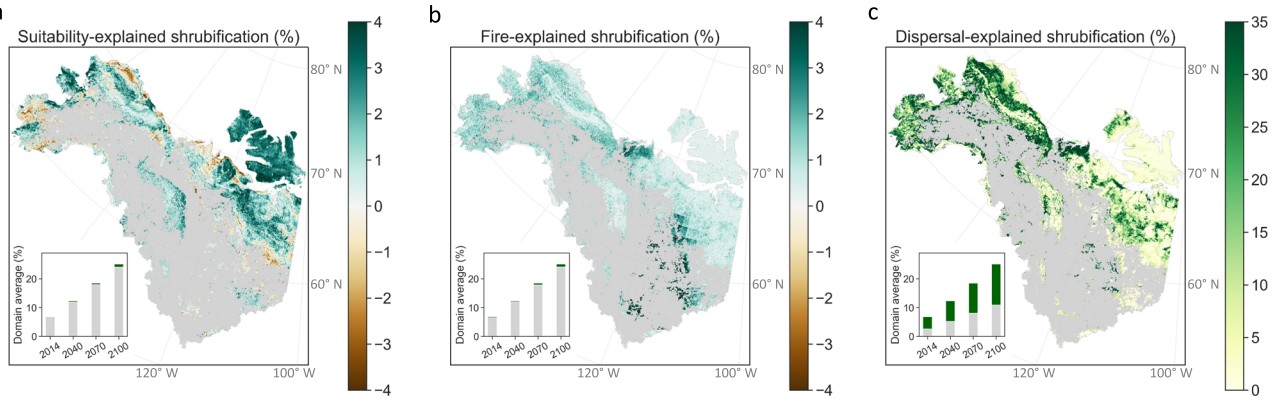

**Fig. 4 Shrub expansion over the 21st century was primarily attributed to seed dispersal.** Shrub expansion driven by (**a**) environmental suitability change from 2014 to 2100, (**b**) projected fire, and (**c**) short- and long-distance seed dispersal. The insets show the domain average of shrub expansion from 2014 to 2100; the gray sections show shrub expansion in a scenario where the corresponding factor was turned off and the green sections represent its contribution.

most of the region due to climate warming, although it decreased in some low elevation areas experiencing reduced snow inputs (Supplementary Fig. 8a). Environmental suitability change had a spatially heterogeneous impact on shrub expansion, i.e., increasing in most areas but decreasing in parts of southwestern Alaska, the North slope, and northern Canada. Given the low sensitivity of shrub expansion to environmental suitability (Fig. 2a) and the spatial compensation, the net result of environmental suitability changes was small averaged across the region (~1%) by 2100. Fires burned 3.2% of the area during 1984–2014, which was projected to increase to 7% during 2070–2100 based on the CMIP6 ensemble average. Burn area in CMIP6 models was mostly concentrated in the southeast (Supplementary Fig. 8b), which had a limited impact on shrub expansion due to low initial shrub cover and thus seed-arrival probability in that region (Supplementary Fig. 8c). In most of the tundra in Alaska and northern Canada, the burn area was projected to be less than 3% until the end of the 21st century. As a result, although areas disturbed by fire were found more likely to experience shrub

expansion (Fig. 2), projected fire only contributed one percentage point out of the 25% shrub expansion by 2100, equivalent to the impact of projected environmental suitability change (Fig. 4b). The spatial pattern of seed-arrival probability mostly followed existing shrub cover, i.e., high in the majority of Alaska and middle of the Northwest Territories in Canada, and low in coastal regions of Alaska, and southeast and north of the Northwest Territories (Supplementary Fig. 8c). Dispersal largely explained shrub expansion in these regions (Fig. 4c). Notably, although the Brooks Range had moderate seed-arrival probability (Supplementary Fig. 8c), shrubs were found unlikely to expand into this region (Fig. 3) due to the limitation of low environmental suitability (Fig. 1c and Supplementary Fig. 8), highlighting the compound impact of environmental suitability and seed dispersal. Across the domain, seed dispersal explained 14% out of the 25% shrubified tundra from 1984 to 2100 (Fig. 4c). Given the dominant control of seed dispersal on the spatial pattern of shrub expansion, omitting dispersal likely leads to mis-represented shrub cover change.

## Discussion

Climate warming has made the Arctic tundra substantially more suitable for shrubs over recent decades. However, we demonstrate that more suitable areas do not necessarily experience more extensive shrub expansion, which, instead, is found in areas close to existing shrub patches and/or disturbed by fire. In contrast to previous findings that suggest a stronger limitation of environmental suitability than seed dispersal over the past millennia[41], the results here indicate dispersal processes limit shrub expansion over recent decades. Our findings provide observational evidence for the importance of seed dispersal in Arctic shrub expansion under rapid warming as the ecosystem deviates from its historical equilibrium. The fact that shrubs did not expand into all suitable areas implies shrub establishment might not have kept up with the pace of recent climate change. Although a high rate of environmental suitability change under the RCP8.5 scenario was used for prediction, in a contrasting scenario where environmental suitability is kept the same as in 2014 through 2100, shrub cover is still predicted to substantially increase across the domain (gray bars in Fig. 4a). Therefore, shrubs will likely continue to expand across the Arctic tundra due to lagged response, even under a net-zero emission scenario, where global warming will be limited to 1.5 °C by 2050 and stabilized by 2100[59].

Complex ecosystem processes introduce uncertainties in predicted environmental suitability, identified shrub expansion, and the relationship to seed dispersal and fire disturbance. Uncertainties related to future environmental suitability can be influenced by future bioclimatic conditions exceeding the historical ranges used to establish their relationships with environmental suitability (Supplementary Fig. 9). For example, nutrient availability could increase much faster with temperature in a warmer climate due to an exponential increase of N mineralization rate and deepening active layer[45]. Thus, the data-driven environmental suitability model trained using historical data could underestimate future environmental suitability. The nonlinear impacts of bioclimatic conditions on environmental suitability (Supplementary Figs. 2 and 3) should also be interpreted as specific to the domain configuration and are subject to uncertainty as the climate shifts beyond the historical regime. Likewise, seed production and dispersal could also deviate from historical regimes due to biotic and abiotic interactions[60,61]. For example, a recent study suggested declined population of animals as dispersal vectors likely further limits long-distance dispersal of plants under future climate[62], thus leading to underestimated dispersal limitation relying on empirical relationships. However, mechanistic models could contribute to addressing these uncertainties. Shrub expansion was identified based on remotely sensed shrub dominance at a 30 m scale and over 30 years, which is subject to land-cover classification errors especially with coexistence of multiple growth forms[3]. Notably, shrub expansion detected at a 30 m resolution may not precisely distinguish the underlying causes of seed dispersal from increased coverage of preexisting shrubs due to enhanced growth or new establishment from very local dispersal (within the 30 m pixel)[3,9]. However, because shrub growth and local seed production are expected to be controlled by environmental suitability, the low impact of environmental suitability supports seed dispersal being the dominant cause of shrub expansion across the domain. Moreover, as dispersal is estimated based on spatial proximity, our results highlight the importance of spatially connected processes. Although seed dispersal is the originating mechanism and has been recognized as a dominant spatial process controlling vegetation range shifts[35–39], the impact of spatial proximity identified here might also be partially attributed to other spatially connected factors, such as active-layer depth, soil thermal-hydro conditions, surface litter, nutrient availability, and herbivore activities[63–66]. These factors may contribute to the spatial connectivity of shrub expansion via rates of seed germination and seedling establishment. However, these factors are unlikely to be the dominant explanation for the identified impact of spatial proximity, as they are partially related to environmental suitability via climate and topographic conditions, and they tend to exhibit smaller spatial ranges than those identified for long-distance dispersal (~40 km). Furthermore, because the remotely sensed land cover that we used cannot distinguish different shrub species while dispersal influences the expansion of each single species, the results based on the aggregation of all shrub species likely overestimate spatial proximity, thus providing conservative estimates of dispersal limitation. Field surveys and measurements are required to investigate the confounding roles of these spatial processes.

Although fires can either enhance or inhibit plant regeneration depending on local soil and climate conditions[25], our results suggest fire enhances shrub expansion where it does occur, consistent with paleoecological studies[44] and model simulations[45,67] across a large scale. The strong compounding effect of fire and seed dispersal on shrub expansion (Fig. 2) highlights that fire promotes shrub expansion especially at locations close to pre-existing shrub patches, where seeds are more likely to arrive and establish after fire. Because the impact of dispersal can be attenuated by competition with preexisting species such as long-lived perennials in the tundra, the enhanced impact of dispersal by fire could be partially attributable to lowered competition through removal of preexisting species. Because fire is projected to be rare in the Arctic tundra based on climate models, we find it only marginally contributes to shrub expansion by 2100. However, a recent study suggests lightning in Arctic tundra, the dominant source of burning[68], will significantly increase to a rate similar to that in boreal forests[16]. Lightning-driven fire increases could trigger positive vegetation-fire feedbacks, leading to twofold more burn area by 2100 than the ensemble average of CMIP6 models (Supplementary Fig. 10)[16,69]. Therefore, fire likely exerts greater impacts on shrub expansion compared to the estimates here when considering these positive feedbacks, though further investigation is required to constrain the large uncertainty (Supplementary Fig. 10). As post-fire regeneration strongly controls how much fire-induced carbon loss is attenuated[17], future work on the strength and spatial heterogeneity of the feedback between fire and shrub expansion will contribute to a better assessment of the carbon budget in Arctic tundra.

Our results highlight that predicting shrub expansion cannot be based on climate alone. Models that do not account for fire disturbance and seed dispersal may misrepresent future shrub cover. In Earth system models, seed production and dispersal have been recognized as the most under-developed vegetation demographic processes[70]. Representing seed dispersal, especially over long distances, requires seed transport across spatially discretized grids, which does not exist in most land models. Improved representation of seed dispersal therefore could contribute to better prediction of vegetation shifts. In addition to the factors investigated here, shrub expansion is also modulated by species competition for water, nutrients, and light[71–73]. Recent observational evidence suggests climate change can result in different competitive abilities across species due to divergent shifts of plant functional traits in Arctic tundra[74–76], highlighting the potential of employing dynamic vegetation models that explicitly represent competition. These findings motivate improving process-based representations of seed dispersal, fire disturbance, and species competition in dynamic vegetation models as a fundamental component to better prediction of Arctic shrub change and corresponding climate feedbacks.

## Methods

**Datasets**. Shrub expansion was identified based on the Landsat-derived product of annual dominant land cover across ABoVE core domain from 1984 to 2014[77]. The dataset provides annual dominant plant functional type at a 30 m resolution derived from Landsat surface reflectance, very high-resolution imagery, and field photography across the ABoVE domain. We focused on pixels dominated by shrubs and non-woody species, i.e., excluding boreal forests. Pixels consistently classified as shrublands during 1984–1986 and 2012–2014 were considered as shrub cover in 1984 and 2014, minimizing the uncertainty of noise in the annual time series of land-cover types. New shrub area was identified as pixels that had been dominated by non-woody species in 1984 and became dominated by shrubs in 2014. We used climate and topographic conditions to estimate environmental suitability for shrublands. The climate conditions listed in Supplementary Table 1 came from ClimateNA[78], a product locally downscaled for North America at a 4 km resolution. The historical data (1955–2014) was downscaled from the gridded Climatic Research Unit Time-series data version 4.02 (CRU TS4.02), and the projected data (2014–2100) was downscaled from CMIP5 under the RCP8.5 scenario, which is broadly consistent with recent trends of global carbon emissions[79]. The elevation data were obtained from the Advanced Spaceborne Thermal Emission and Reflection Radiometer (ASTER) Global Digital Elevation Model Version 3 with a 30 m resolution[80]. Slope, aspect, and the topographic wetness index were derived from elevation using a terrain analysis software RichDEM[81]. Fire occurrence during 1985–2009 was identified using the annual product of differenced Normalized Burned Ratio (dNBR) at a 30 m resolution[82], where the perimeters came from the Alaskan Interagency Coordination Center and the Natural Resources Canada fire occurrence datasets. Only fires that occurred at least 5 years prior to 2014 were considered to allow vegetation recovery. Burn area during 2015–2100 was obtained from CMIP6 projections under a SSP585 scenario. Datasets at coarse resolutions (climate and projected burn area) were resampled to a 30 m resolution using the nearest neighbor method.

**Estimation of environmental suitability**. We considered 23 bioclimatic variables (Supplementary Table 1) and 4 topographic conditions, i.e., elevation, slope, aspect, and topographic wetness index. To reduce the risk of overfitting, we identified the most informative variables based on the variance inflation factor, which measures the multicollinearity among the explanatory variables. Starting from all 27 variables, we excluded the variable with the highest variance inflation factor, i.e., the variable that can be best represented by a linear combination of other variables, one at a time, until the variance inflation factors of all variables are below the commonly used threshold of five[83]. This procedure ensured that the identified variables are most statistically informative in representing the bioclimatic and topographic conditions across the domain. Based on the identified variables, we applied ten species distribution models to estimate whether a pixel was shrubland. The models include generalized linear model, generalized additive model, boosted regression trees, classification tree analysis, artificial neural network, surface range envelope, flexible discriminant analysis, multiple adaptive regression splines, random forest, and maximum entropy, all applied using the *biomod2*[84] software in R[85]. Due to the large computation load, we trained each model using 5% of the pixels randomly selected within the target area, including both shrub and non-shrub pixels. The model accuracies were evaluated using all pixels across the entire domain. The random forest model had the highest accuracy based on the true skill statistic and the area under the receiver operating characteristic curve. Therefore, the environmental suitability, i.e., the probability of a given 30 m pixel being shrubland given its bioclimatic and topographic conditions, was calculated using only a random forest model. We assumed a relatively stable climate prior to 1984. Thus, the average bioclimatic conditions during 1955–1984 were used to train the random forest model and assess environmental suitability in 1984. Environmental suitability in 2014, 2040, 2070, and 2100 was estimated by replacing the bioclimatic conditions to the averages over the previous 30 years, respectively. We evaluated the relative importance of each variable in explaining environmental suitability. We also analyzed the response curve of environmental suitability to the variation of each variable, and the response surfaces to the covariation of the most important three variables, while setting other variables as the domain average.

**Seed-arrival probability**. The impact of seed dispersal was quantified using the probability of seed arrival at a given location, calculated using kernel convolution over the spatial pattern of shrublands. The following exponential power kernel was used to describe the relationship between seed-arrival probability and distance to parent shrub patches.

$$k(x_i) = \frac{b}{2\pi a^2 \Gamma(2/b)} \exp\left(-\left(\frac{x_i}{a}\right)^b\right) \quad (1)$$

where $x_i$ is the distance to the $i$th shrubland pixel within a maximum range, which is considered as the distance where the kernel function first falls below $10^{-9}$; $a$ and $b$ are the range and shape parameters, respectively. Large $a$ represents high seed-arrival probability from distant parent shrub patches and vice versa. Large $b$ denotes a fast decay rate of seed-arrival probability with distance and vice versa. The exponential power kernel is a generalized form of the Gaussian ($b = 2$),

exponential ($b = 1$), and fat-tailed ($b = 0.5$) kernels, and has been widely used in literature[86]. The seed-arrival probability of a given location (**s**) is calculated as follows:

$$p(\mathbf{s}) = \frac{1}{P_{\max}} \sum_{i=1}^{N} k(x_i) \Delta x \quad (2)$$

where $N$ is the total number of shrub pixels within the maximum range; $\Delta x = 30$ m is the width of a pixel; and $P_{\max}$ is the normalization factor such that $p(\mathbf{s}) = 1$ when the location is completely surrounded by shrublands within the maximum range. The seed-arrival probability $p(\mathbf{s})$ measures the spatial proximity to existing shrublands. Assuming the same seed production of all shrublands, the seed arrival probability $p(\mathbf{s})$ is also proportional to the expectation of the arriving seed amount. Based on the shrub cover in 1984, we calculated the seed-arrival probability during 1984–2014 using the above-described algorithm implemented in the multidimensional image processing software of *scipy.ndimage*[87] in Python. As seeds can arrive via multiple dispersal vectors, the seed-arrival probability results from the integral of multiple dispersal kernels with distinct ranges and shapes[32]. To parsimoniously account for various dispersal vectors, we considered the integral of a short-distance dispersal kernel and a long-distance dispersal kernel. For short-distance dispersal, we evaluated all combinations of $100\,\text{m} \le a \le 1000\,\text{m}$ with an interval of 100 m and $0.5 \le b \le 2.5$ with an interval of 0.5. For long-distance dispersal, we evaluated all combinations of $1\,\text{km} < a \le 60\,\text{km}$ with an interval of 2 km and $0.5 \le b \le 2.5$ with an interval of 0.5. Using a larger interval of 2 km facilitates optimization efficiency for the long-distance dispersal kernel. We identified the parameters that resulted in the best 5% accuracy in estimating shrub expansion during 1984–2014. The uncertainty of estimated shrub expansion sensitivity using different kernel parameters across the best 5% was quantified. The relative weights (sensitivities) of the two kernels were identified as those that best explain shrub expansion patterns, using Eqs. (3) and (4). The spatial pattern of the combined kernel density (Supplementary Fig. 7) shows an estimate proportional to seed-arrival probability resulting from both short- and long-distance dispersal.

**Sensitivity of observed shrub expansion to control factors**. Observed shrub expansion was quantified as the fraction of 30-m pixels that were not identified as shrublands in 1984, i.e., non-shrub tundra, but became shrublands in 2014 within each 4 km by 4 km gridcell. Environmental suitability and seed-arrival probabilities through short- and long-distance dispersal were aggregated by average to a 4-km scale and used to explain the spatial pattern of shrub expansion using the following multivariate linear regression.

$$y(\mathbf{s}) = a_0 + a_1 \text{ES}(\mathbf{s}) + a_2 \text{LD}(\mathbf{s}) + a_3 \text{SD}(\mathbf{s}) + \delta(\mathbf{s}) \quad (3)$$

$y(\mathbf{s})$ is the new shrub area; $\text{ES}(\mathbf{s}), \text{LD}(\mathbf{s}), \text{SD}(\mathbf{s})$ are the z-scores of environmental suitability, long-distance dispersal, and short-distance dispersal at location **s**, respectively; and $\delta(\mathbf{s})$ is the noise. The sensitivities $a_1, a_2, a_3$ were estimated for gridcells with and without fire occurrence, respectively. The 95% confidence intervals of the sensitivities were estimated.

In addition to the considered explanatory variables, shrub expansion may also be influenced by other unconsidered confounding factors that lead to a spatial correlation pattern independent from that induced by dispersal. To account for such spatial correlation, we also conducted a spatial regression, i.e.,

$$y(\mathbf{s}) = b_0 + b_1 \text{ES}(\mathbf{s}) + b_2 \text{LD}(\mathbf{s}) + b_3 \text{SD}(\mathbf{s}) + w(\mathbf{s}) + \sigma(\mathbf{s}) \quad (4)$$

where $w(\mathbf{s})$ represents a spatial correlation structure of shrub expansion following a stochastic Gaussian process, which has a zero-mean and is independent from the considered explanatory variables; $\sigma(\mathbf{s})$ is an uncorrelated error. The sensitivities to the environmental suitability and dispersal $b_1, b_2, b_3$ were jointly estimated with $w(\mathbf{s})$ and $\sigma(\mathbf{s})$ using the *spBayes* software[88] in R[89], for grid cells with and without fire occurrence, respectively. An exponential covariance model and the following prior distributions were used based on empirical variogram[90]: $\phi \sim$ Uniform (30 m, 2000 m), $\sigma^2 \sim$ Inverse Gamma (2, 0.05), and $\tau^2 \sim$ Inverse Gamma (2, 0.05), where the $\phi, \sigma^2, \tau^2$ are the range parameter, covariance and nugget effects. Due to the large computing load, the spatial regression was trained using randomly selected 5% of the 4 km pixels and tested on another randomly selected independent set with the same size. The correlation $r$ was estimated for the test set. Sensitivities were calculated as the mean of 5000 posterior samples after convergence (1000 samples).

**Prediction of shrub expansion by 2100**. Based on the estimated empirical relationships, we predicted shrub expansion by 2040, 2070, and 2100. For each 30-year period, environmental suitability was estimated using topographic conditions and the averages of projected bioclimatic conditions. Seed-arrival probability was estimated using shrub cover at the start of the period. The fire was assigned for each 30 m pixel with a probability represented by the cumulative burn area fraction, ensuring the aggregation from a 30 m scale consistent with the climate model projection at a coarser scale. Each non-shrub pixel at the start of the 30-year period was changed to shrubland at the end with a probability calculated using the estimated sensitivities to its environmental suitability, seed-arrival probability, and fire occurrence. We further quantified the uncertainty of projected shrub expansion

due to the uncertainty in the estimated sensitivities. Instead of a computationally expensive bootstrapping approach, we used the lower and upper boundaries of the 95% confidence intervals for all the regression coefficients in each 30-year period, which provided an overestimate of the uncertainty range of projected shrub expansion. To disentangle the impacts of environmental suitability change, seed dispersal, and fire on the projected shrub expansion, we used synthetic scenarios where each of the three factors was turned off, i.e., environmental suitability kept the same as in 2014, zero seed-arrival probability, and no fire occurrence, respectively. The difference between the synthetic scenarios and the actual projection illustrated the contribution of the corresponding factor on the projected shrub expansion. To diagnose potential bias and spatial patterns of predicted shrub expansion using the environmental suitability-based approach in previous studies, the shrub expansion by 2100 predicted here was also compared to the prediction without considering dispersal and fire, i.e., by applying 2100 environmental suitability to the relationship established between shrub presence and environmental suitability alone in 1984.

**Reporting summary**. Further information on research design is available in the Nature Research Reporting Summary linked to this article.

## Data availability

All datasets used in this study are publicly available. The annual land-cover product is available at https://daac.ornl.gov/ABOVE/guides/Annual_Landcover_ABoVE.html. The historical and projected climate conditions and the application to downscale (ClimateNA) is available at https://climatena.ca/. The ASTER DEM is available at https://lpdaac.usgs.gov/products/astgtmv003/. The dNBR product is available at https://daac.ornl.gov/cgi-bin/dsviewer.pl?ds_id=1564. Processed data used to produce the main figures are available at https://doi.org/10.6084/m9.figshare.20097104.v1[91].

## Code availability

The source code used to calculate environmental suitability and seed dispersal is publicly available at https://github.com/YanlanLiu/arctic_shrub_expansion[92].

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

## Acknowledgements

This research was supported by the U.S. Department of Energy, Office of Science, Office of Biological and Environmental Research under Contract No. DEAC02-05CH11231 to Lawrence Berkeley National Laboratory as part of the Next-Generation Ecosystem Experiments in the Arctic (NGEE-Arctic) project. T.F.K. acknowledges support from a NASA Carbon Cycle Science Award 80NSSC21K1705. T.F.K. and Q.Z. acknowledge support from the RUBISCO SFA, which is sponsored by the Regional and Global Model Analysis (RGMA) Program in the Climate and Environmental Sciences Division (CESD) of the Office of Biological and Environmental Research (BER) in the U.S. Department of Energy (DOE) Office of Science. J.A.H. is supported as part of the Energy Exascale Earth System Model (E3SM) project, funded by the U.S. Department of Energy, Office of Science, Office of Biological and Environmental Research (BER). The authors acknowledge the NASA ABoVE team for making the remote sensing products publicly available.

## Author contributions

Y.L. and M.S.T. conceived the study. Y.L. designed the analyses with inputs from W.J.R., T.F.K., Z.A.M., J.A.H., and M.S.T. Q.Z. processed the CMIP6 fire projection data. Y.L. performed the analyses. All authors contributed to interpreting the results. Y.L. wrote the manuscript with inputs from all authors.

## Competing interests

The authors declare no competing interests.
