## [Peer Review File · Nature Communications]

Peer review comments, first round review –

Reviewer #1 (Remarks to the Author):

This is an excellent manuscript describing excellent and original science that is of considerable importance in significantly advancing our understanding of the controls on one of the largest and most visible ecosystem changes that has occurred over the past few decades. The greening of the arctic is one of the most important and clear ecosystem changes that we are seeing worldwide, and has been apparent in the satellite record since the 1980s. However, its main cause (that of “shrubification”) has generally been linked to climate change. This manuscript significantly changes that view by showing that environmental suitability (essentially “warming”) has had a lesser control on shrub expansion than fire and seed dispersal. Furthermore, given fire is increasing in frequency in tundra ecosystems, this adds a substantially important understanding to how shrubification, and hence arctic greening, may proceed in the future. So the work is important because it deals with a major ecosystem change that is happening now, and is of considerable significance because it substantially improves how we understanding that change.

The work is robust and the manuscript is well written. The methods are sound and well described. The language also makes it accessible to a broad readership as is appropriate for publication in Nature Comms. Largely my comments are minor, or where more important, I believe they can be dealt with relatively easily in the manuscript and do not distract away from the manuscript’s and science’s quality.

My main point would be that there is little discussion of the fact that shrubs also spread by vegetative growth. This is often the way for expansion of shrub patches which has been shown to occur as an important part of shrubification. It is not clear to me how that is taken account for very local shrub expansion, or how it might apply for long-range dispersal where a shrub establishes somewhere a long way from other shrubs but then vegetative growth drives its local expansion. The implications of this needs raising in the introduction and discussion. Does it modify the way the results need interpreting? Are there caveats around this, or does the way the model works means it is not that important?

Related to that there is no mention that establishment from seed is highly challenging in tundra landscapes dominated by long-lived perennials where there is much pre-emption of space. While this could suggest a limitation to the important of seed dispersal in tundra, it potentially also lends weight to the importance of fire since fire opens up safe sites for seedling establishment.

The term “suitability” is used throughout to describe the environmental suitability for shrub cover. I think this would be better being called “environment” since to me suitability also includes things like existence of safe sites for seed establishment. OK, that would also be true for “environment” but it will help to get across the idea that this is challenging the notion that shrubification is largely a climate driven process. “Suitability” is too vague I think.

Line 12: delete “large” because there is great uncertainty as to the consequences of shrubification to these processes.

Line 17: To help accessibility for a general audience, change “biotic-abiotic interactions and non-stationary relationships”. This is unclear and will not help attract a boarder interest reader to the

paper. Similarly line 54 “ignores transient responses...” – explain more of what you mean.
Line 19: “Shrub expansion has not been controlled by environmental suitability”. This is really big news and I wonder if it can be added to the title?
Line 188: Some citations needed here.

Reviewer #2 (Remarks to the Author):

This is an interesting paper that explores the role of dispersal and establishment probabilities (via fire) on the spread of shrub in the Arctic under climate change. It suggests that without understanding dispersal, we will be unable to predict land cover change accurately, and this has consequences for feedback processes, such as albedo. It has long been known that dispersal and population processes are an important constraint on responses to climate change, and that needs to be considered in assessing drivers of species' ranges¹⁻³, but, as the authors argue, this is still rarely analysed. There is much research on estimating demographic parameters from environmental data⁴ and new software for estimating dispersal in SDMs⁵. The paper presents an interesting approach, but I feel there are some issues which need to be addressed.

i. My primary concern is the assumption that spatial signals in the data are due to dispersal patterns. Spatial auto-correlation is a major issue in fitting Species Distribution Models (SDM) as both distribution and environmental data are spatially structured^{6,7}. There seems to be no analysis of spatial autocorrelation in fitting the initial environmental models, which can cause errors⁷ but also leaves any spatial signal to be picked up by the dispersal kernels. Thus the analysis may artificially bias the estimate of the importance of dispersal. I would emphasise that this is my take on reading the paper and it would be good to see a response and additional analysis to address my concern.

ii. I have another concern that 2 dispersal kernels are used. It is not clear why this is done, other than the vague statement that there may be multiple dispersal mechanisms. An exponential power is used for both. An advantage of the exponential power is that it can give a fat-tailed kernel (for $b < 1$). Note 'fat' not 'flat' tailed. The fat tail allows long distance dispersal, so why have two functions? This smacks of over-fitting and not really understanding the dispersal process. There is the added concern that combining 2 kernels may mean the resultant kernel is not a true probability density function as the integral may > 1 , so the actual amount of dispersal varies depending on kernel parameters.

iii. Climate change can affect dispersal directly, e.g.⁸, so future dispersal may be different to current dispersal. This is not something the authors can address as they are not using mechanistic dispersal models. But it may be worth mentioning as a caveat.

iv. It is not clear to me if multiple kernels with different parameter values (i.e. the best 5%) were used, and if so, how that variety is captured.

v. The problem with fitting a Random Forest model is illustrated in Extended Data Fig 2, whereby there are complex relationships with environmental drivers. This is especially true for DD5 and PAS. It seems unlikely that these complex relationships are true. For example, for DD5, suitability is higher at ca 500 than at ca 700, but is high again at ca 800. I think some discussion is needed. Collinearity in drivers has been considered, so I assume that is not an issue. Maybe there is an issue with fitting one

model across the multiple species represented by the shrub communities (see below)?

vi. The shrublands are not well described ecologically. What species do they comprise, what are the life histories of the main species, and does species composition vary greatly across the region? The suitability and dispersal approaches would generally be used for a single species (e.g. SDMs are fitted species by species). While fitting overall models is OK, there are errors, uncertainties and biases in doing so. The biggest bias may arise if 'shrubland' means a very different community depending on location. So, a better explanation of what these ecosystems comprise would be useful, as well as a discussion of these issues I have noted.

vii. It is not clear to me that the shrubland cover in 2014 is simply the cover in 1984 + new pixels. That is, are there areas with shrubland in 1984 that did not have shrubland in 2014? The NW region in particular seems to have a loss of shrubland between 1984 and 2014 (Fig. 1). It is hard to work out what is happening, but this may be an issue

viii. What are the grey areas on the maps? This needs explanation in the legends.

ix. There is no discussion of errors and uncertainties in classifying pixels as shrubland or other.

x. Climate data are at 4km, but the pixels were 30m. What downscaling procedure was used?

xi. The future climate was based on a single model and used RCP 8.5. Climate models differ in their projections for each RCP, and 8.5 is the most extreme climate scenario. The RCPs are projections of possible futures, so it is misleading to use statements like "A quarter of non-woody tundra in 1984 will be colonized by shrubs by 2100". Some discussion about the choices made here & caveats concerning other RCPs would be useful.

1 Engler, R., Hordijk, W. & Guisan, A. The MIGCLIM R package - seamless integration of dispersal constraints into projections of species distribution models. *Ecography* 35, 872-878, doi:10.1111/j.1600-0587.2012.07608.x (2012).

2 Brooker, R. W., Travis, J. M. J., Clark, E. J. & Dytham, C. Modelling species' range shifts in a changing climate: The impacts of biotic interactions, dispersal distance and the rate of climate change. *Journal of Theoretical Biology* 245, 59-65, doi:10.1016/j.jtbi.2006.09.033 (2007).

3 Schurr, F. M. et al. How to understand species' niches and range dynamics: a demographic research agenda for biogeography. *J. Biogeogr.* 39, 2146-2162, doi:10.1111/j.1365-2699.2012.02737.x (2012).

4 Holden, M. H. et al. Assessing the accuracy of density-independent demographic models for predicting species ranges. *Ecography* 44, 345-357, doi:10.1111/ecog.05250 (2021).

5 Shipley, B. R. et al. megaSDM: integrating dispersal and time-step analyses into species distribution models. *Ecography* n/a, doi:https://doi.org/10.1111/ecog.05450.

6 Mielke, K. P. et al. Disentangling drivers of spatial autocorrelation in species distribution models. *Ecography* 43, 1741-1751, doi:https://doi.org/10.1111/ecog.05134 (2020).

7 Lany, N. K., Zarnetske, P. L., Finley, A. O. & McCullough, D. G. Complementary strengths of spatially-explicit and multi-species distribution models. *Ecography* 43, 456-466, doi:https://doi.org/10.1111/ecog.04728 (2020).

8 Bullock, J. M. et al. Modelling spread of British wind-dispersed plants under future wind speeds in a changing climate. *Journal of Ecology* 100, 104-115, doi:10.1111/j.1365-2745.2011.01910.x (2012).

Reviewer #3 (Remarks to the Author):

Main issues:

This is an excellent study and a fantastic presentation overall. My only substantial concern is that future fire activity is substantially misrepresented. Lines 221 – 222 and 288-290 claim that in the majority of Alaska fire was projected to be near zero until the end of 2100... Fire in Alaskan tundra has been shown to be a rather common and extensive disturbance (see French et al 2015, Higuera et al 2011, He et al. 2019). And as the authors state, it is widely expected to increase by 2100 almost under any scenario (and definitely under the RCP8.5 considered in this study). It is definitely not occurring at the levels observed in the boreal forests but it is common particularly for the study area of this present project.

I don't believe that this will invalidate the major findings of this research that points to seed dispersal and fire occurrence as the dominant factors of shrubification compared to environmental suitability alone. However, it considerably increases the uncertainty of future projections considering that fire is poorly represented.

Specific suggestions:

Lines 65-70: I believe this section is meant to indicate that in the Arctic seed dispersal is likely to occur over large distances. But I am not entirely sure. It might be helpful if the authors states explicitly their point here.

Line 72: Depending on definition of "historic", this may not be entirely accurate. French et al 2015 show that over the past 80 years, large fires have been rather common in Alaska. Granted Alaska is substantially more impacted by fire than any other Arctic region, however, considering that this study focuses on Alaska and Westerns Canada, it should be acknowledged.

Lines 120-123: If I understand the authors' intent correctly, the Pearson's r here is used to show how closely the model based entirely on environmental suitability can describe shrub distribution (essentially, how much variance it accounts for). However, the way the text reads it appears that the authors use this metric as a separate validation of their model's performance which wouldn't make a whole lot of sense considering that the model is derived based on the shrub distribution. Perhaps a slight clarification of the text would be beneficial.

Fig 1: Please add the description units of suitability and the range to the figure caption.

Line 210: "... turning-off on factor..." should probably be "...turning off one factor..."

Authors: We thank the reviewer/s for the constructive and supportive comments. To address the comments, we have added a new analysis that accounts for spatial correlation in estimating observed shrub expansion. We also incorporated multiple clarifications and discussion points, which we believe have strengthened the manuscript. The reply to each specific comment (in blue) is described below (in black). The line numbers correspond to those in the revised manuscript. Figures included in this reply but not in the manuscript are numbered as Fig. Rxx.

REVIEWER COMMENTS

Reviewer #1 (Remarks to the Author):

This is an excellent manuscript describing excellent and original science that is of considerable importance in significantly advancing our understanding of the controls on one of the largest and most visible ecosystem changes that has occurred over the past few decades. The greening of the arctic is one of the most important and clear ecosystem changes that we are seeing worldwide, and has been apparent in the satellite record since the 1980s. However, its main cause (that of “shrubification”) has generally been linked to climate change. This manuscript significantly changes that view by showing that environmental suitability (essentially “warming”) has had a lesser control on shrub expansion than fire and seed dispersal. Furthermore, given fire is increasing in frequency in tundra ecosystems, this adds a substantially important understanding to how shrubification, and hence arctic greening, may proceed in the future. So the work is important because it deals with a major ecosystem change that is happening now, and is of considerable significance because it substantially improves how we understanding that change.

The work is robust and the manuscript is well written. The methods are sound and well described. The language also makes it accessible to a broad readership as is appropriate for publication in Nature Comms. Largely my comments are minor, or where more important, I believe they can be dealt with relatively easily in the manuscript and do not distract away from the manuscript’s and science’s quality.

Authors: We thank the reviewer for the supportive comment on the scientific merit of the manuscript. Our response and corresponding revisions regarding the following comments are described below.

My main point would be that there is little discussion of the fact that shrubs also spread by vegetative growth. This is often the way for expansion of shrub patches which has been shown to occur as an important part of shrubification. It is not clear to me how that is taken account for very local shrub expansion, or how it might apply for long-range dispersal where a shrub establishes somewhere a long way from other shrubs but then vegetative growth drives its local expansion. The implications of this needs raising in the introduction and discussion. Does it modify the way the results need interpreting? Are there caveats around this, or does the way the model works means it is not that important?

Authors: This point is well taken. We agree that shrub expansion can be caused by long-distance dispersal, short-distance dispersal, and growth of pre-existing shrubs. Our approach characterizes both short- and long-distance dispersal. The short-distance dispersal characterizes local expansion across ranges from 30 m to 1 km. The fact short-distance dispersal explains the shrub

expansion pattern highlights the prevalence of short-distance dispersal. However, shrub expansion can also arise from non-dominant shrubs becoming dominant due to growth within a 30 m scale, including both increased biomass/canopy coverage of pre-existing shrub individuals and new establishment from very local dispersal (within the 30 m pixel). While such impact cannot be captured using remote sensing data at a 30 m resolution, we do not expect the uncertainty would qualitatively change the interpretation. Local growth and establishment within a 30 m range are considered less of importance as they heavily rely on environmental suitability, which was found to minimally affect shrub expansion pattern.

To clarify this caveats and why it will not qualitatively change the interpretation of the findings, we added the following in lines 296-299: “Notably, shrub expansion detected at a 30 m resolution may not precisely distinguish the underlying causes of seed dispersal from increased coverage of pre-existing shrubs due to enhanced growth or new establishment from very local dispersal (within the 30 m pixel) (Wang et al. 2020; Myers-Smith et al. 2020). However, because shrub growth and local seed production are expected to be controlled by environmental suitability, the low impact of suitability supports seed dispersal being the dominant cause of shrub expansion across the domain.”

Related to that there is no mention that establishment from seed is highly challenging in tundra landscapes dominated by long-lived perennials where there is much pre-emption of space. While this could suggest a limitation to the importance of seed dispersal in tundra, it potentially also lends weight to the importance of fire since fire opens up safe sites for seedling establishment.

Authors: Thanks for this constructive comment. We agree establishment is affected by competition with preexisting species. Indeed, our results show that shrub expansion is more sensitive to dispersal at locations disturbed by fire (Fig. 2a), partially attributable to reduced competition. To address this point, we added in lines 323-326: “Because the impact of dispersal can be attenuated by competition with preexisting species such as long-lived perennials in tundra, the enhanced impact of dispersal by fire could be partially attributable to lowered competition through removal of pre-existing species. ”

The term “suitability” is used throughout to describe the environmental suitability for shrub cover. I think this would be better being called “environment” since to me suitability also includes things like existence of safe sites for seed establishment. OK, that would also be true for “environment” but it will help to get across the idea that this is challenging the notion that shrubification is largely a climate driven process. “Suitability” is too vague I think.

Authors: We edited “suitability” to “environmental suitability” throughout to clarify the suitability here was estimated using environmental conditions (bioclimatic and topographic conditions).

Line 12: delete “large” because there is great uncertainty as to the consequences of shrubification to these processes.

Authors: We deleted the word “large”. The sentence was edited to “Arctic shrub expansion has been widely reported in recent decades, altering carbon budgets, albedo, and warming rates in high latitudes.”

Line 17: To help accessibility for a general audience, change “biotic-abiotic interactions and

non-stationary relationships”. This is unclear and will not help attract a broader interest reader to the paper. Similarly line 54 “ignores transient responses...” – explain more of what you mean.

Authors: We rephrased “biotic-abiotic interactions and non-stationary relationships” to “demographic processes and non-stationary response of shrubs to changing climate” (lines 17-18).

Line 19: “Shrub expansion has not been controlled by environmental suitability”. This is really big news and I wonder if it can be added to the title?

Authors: Thanks for the comment. We think the term “environmental suitability” requires more context and explanation, as also pointed out by Reviewer 2. To keep the title concise, we keep it as is and highlight the main finding that “shrub expansion has not been controlled by environmental suitability” in the abstract.

Line 188: Some citations needed here.

Authors: We added related citations on shrub loss in the sentence (lines 204-205): “The resulting pattern (Fig. 3a) does not account for shrub loss due to competition, pests, and herbivores (Mack et al. 2011; Ims and Henden 2012; Wang et al. 2020; Yaping Chen, Hu, and Lara 2021), which are beyond the scope of our study.”

Reviewer #2 (Remarks to the Author):

This is an interesting paper that explores the role of dispersal and establishment probabilities (via fire) on the spread of shrub in the Arctic under climate change. It suggests that without understanding dispersal, we will be unable to predict land cover change accurately, and this has consequences for feedback processes, such as albedo. It has long been known that dispersal and population processes are an important constraint on responses to climate change, and that needs to be considered in assessing drivers of species' ranges¹⁻³, but, as the authors argue, this is still rarely analysed. There is much research on estimating demographic parameters from environmental data⁴ and new software for estimating dispersal in SDMs⁵. The paper presents an interesting approach, but I feel there are some issues which need to be addressed.

Authors: We thank the reviewer for the positive comments. Our reply and corresponding revisions to address each of the following specific comments are described below.

i. My primary concern is the assumption that spatial signals in the data are due to dispersal patterns. Spatial auto-correlation is a major issue in fitting Species Distribution Models (SDM) as both distribution and environmental data are spatially structured^{6,7}. There seems to be no analysis of spatial autocorrelation in fitting the initial environmental models, which can cause errors⁷ but also leaves any spatial signal to be picked up by the dispersal kernels. Thus the analysis may artificially bias the estimate of the importance of dispersal. I would emphasise that this is my take on reading the paper and it would be good to see a response and additional analysis to address my concern.

Authors: This point is well taken. We agree spatial autocorrelation can play a role in explaining the spatial pattern of shrub expansion. To address this point, we added analysis using the following spatial regression, as now described in the Methods section (lines 457-473): “In addition to the considered explanatory variables, shrub expansion may also be influenced by other unconsidered confounding factors that lead to a spatial correlation pattern independent from that induced by dispersal. To account for such spatial correlation, we also conducted a spatial regression, i.e.,

$$y(s) = b_0 + b_1ES(s) + b_2LD(s) + b_3SD(s) + w(s) + \sigma(s)$$

where s refers to given gridcell; $y(s)$ is the new shrub area; $ES(s)$, $LD(s)$, $SD(s)$ are the z-scores of environmental suitability, long-distance dispersal, and short-distance dispersal at location s , respectively; $w(s)$ represents a spatial correlation structure of shrub expansion following a stochastic Gaussian process, which has a zero-mean and is independent from the considered explanatory variables; $\sigma(s)$ is uncorrelated error. The sensitivities to the environmental suitability and dispersal b_1 , b_2 , b_3 were jointly estimated with $w(s)$ and $\sigma(s)$ using the *spBayes* software (Finley, Banerjee, and Gelfand 2015) in R (Team and Others 2013), for gridcells with and without fire occurrence, respectively.”

We found slightly improved test accuracies when considering the spatial correlation, i.e., 0.61 to 0.65 for no-fire areas and 0.71 to 0.74 for fire-disturbed areas, respectively. Notably, the sensitivities of shrub expansion to the explanatory variables remain similar (Extended Data Fig. 6). The result suggests residual spatial correlations exist due to unaccounted factors. However, the estimated spatial correlation structure only marginally correlated to the spatial variations of dispersal and environmental suitability, thus leaving their relative impacts on shrub expansion fundamentally unchanged. Given the advantage of being parsimonious and the more direct implication for applying the identified relationships in modeling and observational studies, we

keep the results from linear regression in the main text. However, we added the discussion of the results of the spatial regression analyses in lines 174-178: “We note that the shrub expansion pattern can also be influenced by other factors unaccounted for, leading to a spatial correlation pattern unexplained by the considered covariates (Mielke et al. 2020). Nonetheless, additionally accounting for spatial correlation of shrub expansion patterns using a spatial regression (Eq. 4 in Methods) only slightly improved estimation accuracy but did not fundamentally alter the estimated sensitivities (Extended Data Fig. 6).”

Extended Data Fig. 6 | Sensitivities of shrub expansion to the environmental suitability and dispersal with and without considering spatial correlation. Considering spatial correlation (darker colors) improves the correlation between estimated and observed shrub expansion (r) from 0.61 to 0.65 for areas without fire and from 0.71 to 0.74 for areas with fire, respectively. The relative sensitivities remained similar.

ii. I have another concern that 2 dispersal kernels are used. It is not clear why this is done, other than the vague statement that there may be multiple dispersal mechanisms. An exponential power is used for both. An advantage of the exponential power is that it can give a fat-tailed kernel (for $b < 1$). Note ‘fat’ not ‘flat’ tailed. The fat tail allows long distance dispersal, so why have two functions? This smacks of over-fitting and not really understanding the dispersal process. There is the added concern that combining 2 kernels may mean the resultant kernel is not a true probability density function as the integral may > 1 , so the actual amount of dispersal varies depending on kernel parameters.

Authors: Using two dispersal kernels is a parsimonious surrogate of the “total dispersal kernel” (Rogers et al. 2019), which is a weighted integral of multiple dispersal kernels resulting from dispersal vectors with distinct shapes and ranges. We agree that ideally, one optimized exponential power kernel might be representative for all dispersal vectors. However, using two dispersal kernels allows greater flexibility to capture the integrated results of the total dispersal kernel, with the weights informed by data. Optimizing the kernel parameters is also more computationally feasible by separating the long-distance dispersal kernel from the short-distance dispersal kernel. As the seed arrival probability barely changes within 100 m (the step size of range parameter optimization for short distance dispersal) at long-distance, using a larger step size of 2 km allows us to effectively identify the optimal range parameter without sacrificing accuracy.

We consider using two kernels is unlikely to cause overfitting because (1) the number of data points ($n = 174,308$) is much higher than the number of covariates ($p = 3$), and (2) the resulting sensitivities are significantly different from zero with relatively small uncertainty ranges (Fig. 2a).

The weighted summation of the two kernel densities can be normalized to a probability between 0 and 1 using a constant, as typically done in analyses using total dispersal kernel (Rogers et al. 2019) and other kernel-based methods (Hastie, Friedman, and Tibshirani 2009). However, it is the spatial pattern of the combined kernel density (Extended Data Fig. 8c) rather than the normalization to probability (between 0 to 1) that is a particular concern here, i.e., it does not violate the principle if the summation of both kernels exceeds 1. This is because the sensitivity and thus prediction of shrub expansion are all based on a linear model (Eq. 3). Thus, if normalized from kernel density to probability, the normalization coefficient will be absorbed into the regression coefficients, whereas the sensitivities (Fig. 2) to the z-scores will remain the same.

To clarify these points, we added the following in the Methods section describing the use of two kernels (lines 429-443): “As seeds can arrive via multiple dispersal vectors, the seed arrival probability results from the integral of multiple dispersal kernels with distinct ranges and shapes (Rogers et al. 2019). To parsimoniously account for various dispersal vectors, we considered the integral of a short-distance dispersal kernel and a long-distance dispersal kernel... Using a larger interval of 2 km facilitates optimization efficiency for the long-distance dispersal kernel. The relative weights (sensitivities) of the two kernels were identified as those that best explain shrub expansion patterns, using Eq. (3) and (4). The spatial pattern of the combined kernel density (Extended Data Fig. 7) shows an estimate proportional to seed arrival probability resulting from both short- and long-distance dispersal.”

We corrected “flat-tailed” to “fat-tailed”.

iii. Climate change can affect dispersal directly, e.g.8, so future dispersal may be different to current dispersal. This is not something the authors can address as they are not using mechanistic dispersal models. But it may be worth mentioning as a caveat.

Authors: Thanks for the suggestion. We added acknowledgement on this caveat in the Discussion section (lines 288-293): “Likewise, seed production and dispersal could also deviate from historical regimes due to biotic and abiotic interactions (Engler et al. 2009; Travis et al. 2013). For example, a recent study suggested declined population of animals as dispersal vectors likely further limits long-distance dispersal of plants under future climate (Fricke et al. 2022), thus leading to underestimated dispersal limitation relying on empirical relationships. However, mechanistic models could contribute to addressing these uncertainties.”

iv. It is not clear to me if multiple kernels with different parameter values (i.e. the best 5%) were used, and if so, how that variety is captured.

Authors: Yes, we tested the accuracies in explaining observed shrub expansion using all possible combinations of a short-distance dispersal kernel and a long-distance dispersal kernel within the specified range (lines 433-436 in Methods): “For short-distance dispersal, we evaluated all combinations of $100 \text{ m} \leq a \leq 1000 \text{ m}$ with an interval of 100 m and $0.5 \leq b \leq 2.5$ with an interval of 0.5. For long-distance dispersal, we evaluated all combinations of $1 \text{ km} < a \leq 60 \text{ km}$ with an interval of 2 km and $0.5 \leq b \leq 2.5$ with an interval of 0.5.” The best 5% were chosen. Each combination resulted in slightly varying sensitivities, illustrated by the vertical

black lines in Fig. 2a. In the figure caption, we noted: “Vertical black lines denote the range of the 95% confidence interval of the regression coefficients across optimal dispersal kernel parameters”. Given the relatively small uncertainty ranges due to different parameters, only the ensemble means of the kernel parameters (Extended Data Fig. 5) were used for predictions. To clarify, we added in lines 438-439: “The uncertainty of estimated shrub expansion sensitivity using different kernel parameters across the best 5% was quantified.”

v. The problem with fitting a Random Forest model is illustrated in Extended Data Fig 2, whereby there are complex relationships with environmental drivers. This is especially true for DD5 and PAS. It seems unlikely that these complex relationships are true. For example, for DD5, suitability is higher at ca 500 than at ca 700, but is high again at ca 800. I think some discussion is needed. Collinearity in drivers has been considered, so I assume that is not an issue. Maybe there is an issue with fitting one model across the multiple species represented by the shrub communities (see below)?

Authors: This point is well taken. We agree that the Random Forest model accounts for relationships of suitability with bioclimate and topographic conditions that are more complex than monotonic relationships. The response surfaces (Extended Data Fig. 3) show that while the suitability generally increases with DD5 and PAS, nonmonotonic responses exist within the 2-D domain, and expectedly, even more so in the 7-dimensional domain accounting for all variables. Therefore, the response curve does not show the entire response function, as other factors were held at the domain averages (lines 403-406). The domain average of elevation is around 400 m (annotated using black dashed line below). Based on the response surface, the suitability exhibits a reduction between DD5 of 500 and 700 degree-day (as described in the comment), which, however, is not always the case in lower or higher elevations. The fact that the Random Forest model outperforms many other species distribution models that assume linear or monotonic dependencies (as described in lines 387-392) support the existence of such nonmonotonic response.

Extended Data Fig. 3 | Response surface of suitability to annual degree-days above 5 °C (DD5), annual precipitation as snow (PAS), and elevation.

However, we note that the response curves and surfaces should be interpreted as specific to the domain configuration. We agree that the two reasons described in the comment could contribute to the complex response, i.e., estimating one suitability model for shrublands consisting of multiple species and collinearity (or more generally, the joint distribution) across bioclimatic and topographic variables. To address this point, we added in lines 121-124: “The nonmonotonic responses could be partially attributable to coexistence of multiple shrub species

that have different optimal environmental conditions, and regional collinearity among bioclimatic and topographic conditions that may not be completely disentangled using a data-driven approach.” Regarding variable collinearity, we used variance inflation factors to remove less informative variables with variance inflation factors exceeding the threshold of five (lines 375-382), which, however, does not guarantee the remaining variables are statistically independent. The figures below show the relationships between average elevation within a small bin of DD5 and PAS. Elevation between DD5 of 500 to 700 is much higher than that under DD5 > 800. Because a higher elevation is associated with a lower suitability, the impacts of elevation and DD5 may not be completely disentangled using a data-driven model, thus contributing to the decreased suitability with increasing DD5 within the range. We acknowledged this uncertainty in the Discussion section (lines 285-288): “The nonlinear impacts of bioclimatic conditions on suitability (Extended Data Fig. 2 and 3) should also be interpreted as specific to the domain configuration and are subject to uncertainty as the climate shifts beyond the historical regime.”

Fig. R1 | Elevation averaged within small bins of degree days above 5 °C (DD5) and precipitation as snow (PAS) across the studied domain.

vi. The shrublands are not well described ecologically. What species do they comprise, what are the life histories of the main species, and does species composition vary greatly across the region? The suitability and dispersal approaches would generally be used for a single species (e.g. SDMs are fitted species by species). While fitting overall models is OK, there are errors, uncertainties and biases in doing so. The biggest bias may arise if ‘shrubland’ means a very different community depending on location. So, a better explanation of what these ecosystems comprise would be useful, as well as a discussion of these issues I have noted.

Authors: We added description on the major shrub communities of shrublands in the region in the Introduction (lines 88-91): “Areas classified as shrubland include prostrate dwarf-shrub tundra and erect-shrub tundra (CAVM Team 2003), dominated by species of birch (*Betula* spp.), alder (*Alnus* spp.), willow (*Salix* spp.), and other dwarf evergreen and semi-deciduous shrubs. Field surveys have detected expansion of these shrub communities (Myers-Smith et al. 2011; Lantz, Marsh, and Kokelj 2013; Wilson and Nilsson 2009).” Discussion on the potential bias has also been added in lines 312-317: “Furthermore, because remotely sensed land cover that we used cannot distinguish different shrub species while dispersal influences expansion of each single species, the results based on aggregation of all shrub species likely overestimate spatial proximity, thus providing conservative estimates of dispersal limitation. Field surveys and measurements are required to investigate the confounding roles of these spatial processes.”

vii. It is not clear to me that the shrubland cover in 2014 is simply the cover in 1984 + new pixels. That is, are there areas with shrubland in 1984 that did not have shrubland in 2014? The NW region in particular seems to have a loss of shrubland between 1984 and 2014 (Fig. 1). It is hard to work out what is happening, but this may be an issue

Authors: We agree. We did not estimate shrub cover in 2014 as shrub cover in 1984 plus new shrub area, since there could also be areas covered by shrubs 1984 but not in 2014, i.e., shrub loss areas, in some places. Shrub loss requires further investigation but is beyond the scope of this study. Our focus in this paper was specifically on controls of shrub expansion, i.e., new shrub area. The definition in lines 91-92 states we focused on shrub gain: “Shrub expansion is defined as shrub dominance in tundra originally dominated by non-woody species at a 30 m scale”. Fig. 3 depicts shrub encroachment not total shrub area. We also clarified in lines 205-206 that shrub loss was not investigated: “The resulting pattern (Fig. 3a) does not account for shrub loss due to competition, pests, and herbivores, which are beyond the scope of our study.”

viii. What are the grey areas on the maps? This needs explanation in the legends.

Authors: We added explanation in the figure caption: “The grey areas are dominated by land cover types other than shrubs and non-woody plants and are excluded from the analyses.”

ix. There is no discussion of errors and uncertainties in classifying pixels as shrubland or other.

Authors: We acknowledged the uncertainty on land cover classification errors in the Discussion section (lines 294-296): “Shrub expansion was identified based on remotely-sensed shrub dominance at a 30 m scale and over 30 years, which is subject to land cover classification errors especially with coexistence of multiple growth forms (Wang et al. 2020).

x. Climate data are at 4km, but the pixels were 30m. What downscaling procedure was used?

Authors: The climate data was kept at 4 km and the shrub expansion were quantified at a 4 km scale as the “the fraction of 30 m pixels that were not identified as shrublands in 1984, i.e., non-shrub tundra, but became shrublands in 2014 within each 4 km by 4 km gridcell” (lines 447-449). To clarify, we also added explanation on the “new shrub area” in the captions of Fig. 1 and 2 as “areal fraction at a 4 km scale”.

xi. The future climate was based on a single model and used RCP 8.5. Climate models differ in their projections for each RCP, and 8.5 is the most extreme climate scenario. The RCPs are projections of possible futures, so it is misleading to use statements like “A quarter of non-woody tundra in 1984 will be colonized by shrubs by 2100”. Some discussion about the choices made here & caveats concerning other RCPs would be useful.

Authors: Thanks for the comment. We clarified in the caption of Fig. 3 that “A quarter of non-woody tundra in 1984 will be colonized by shrubs by 2100 based on the climate scenario RCP8.5”. The scenario RCP8.5 was used as the recent trends of global carbon emissions are broadly consistent with this high emission scenario (Schwalm, Glendon, and Duffy 2020) (lines 430-431). Because suitability change has limited impact on shrub expansion, a different climate scenario will only lead to minimal difference in the predicted change. In Fig. 4a, we tested a scenario where suitability was kept unchanged, in contrast to the fast-changing scenario under RCP8.5, the total shrub expansion area was predicted to be similar though with a slightly altered spatial pattern. We clarified this point in lines 273-278: “Although a high rate of environmental

suitability change under the RCP8.5 scenario was used for prediction, in a contrasting scenario where environmental suitability is kept the same as in 2014 through 2100, shrub cover is still predicted to substantially increase across the domain (grey bars in Fig. 4a). Therefore, shrubs will likely continue to expand across the Arctic tundra due to lagged response, even under a net-zero emission scenario, where global warming will be limited to 1.5 °C by 2050 and stabilized by 2100 (IPCC 2018).”

1 Engler, R., Hordijk, W. & Guisan, A. The MIGCLIM R package - seamless integration of dispersal constraints into projections of species distribution models. *Ecography* 35, 872-878, doi:10.1111/j.1600-0587.2012.07608.x (2012).

2 Brooker, R. W., Travis, J. M. J., Clark, E. J. & Dytham, C. Modelling species' range shifts in a changing climate: The impacts of biotic interactions, dispersal distance and the rate of climate change. *Journal of Theoretical Biology* 245, 59-65, doi:10.1016/j.jtbi.2006.09.033 (2007).

3 Schurr, F. M. et al. How to understand species' niches and range dynamics: a demographic research agenda for biogeography. *J. Biogeogr.* 39, 2146-2162, doi:10.1111/j.1365-2699.2012.02737.x (2012).

4 Holden, M. H. et al. Assessing the accuracy of density-independent demographic models for predicting species ranges. *Ecography* 44, 345-357, doi:10.1111/ecog.05250 (2021).

5 Shipley, B. R. et al. megaSDM: integrating dispersal and time-step analyses into species distribution models. *Ecography* n/a, doi:<https://doi.org/10.1111/ecog.05450>.

6 Mielke, K. P. et al. Disentangling drivers of spatial autocorrelation in species distribution models. *Ecography* 43, 1741-1751, doi:<https://doi.org/10.1111/ecog.05134> (2020).

7 Lany, N. K., Zarnetske, P. L., Finley, A. O. & McCullough, D. G. Complementary strengths of spatially-explicit and multi-species distribution models. *Ecography* 43, 456-466, doi:<https://doi.org/10.1111/ecog.04728> (2020).

8 Bullock, J. M. et al. Modelling spread of British wind-dispersed plants under future wind speeds in a changing climate. *Journal of Ecology* 100, 104-115, doi:10.1111/j.1365-2745.2011.01910.x (2012).

Authors: We thank the reviewer for sharing the references. We have incorporated 4 out of 8 of the references as appropriate in lines 61-62: “Seed dispersal has been investigated in previous studies to estimate species range shifts (Bullock et al. 2012; Rogers et al. 2019; Shipley et al. 2022)...” and in lines 175-177: “We note that shrub expansion pattern can also be influenced by other unaccounted factors leading to a spatial correlation pattern unexplained by the considered covariates(Mielke et al. 2020)”

Reviewer #3 (Remarks to the Author):

Main issues:

This is an excellent study and a fantastic presentation overall. My only substantial concern is that future fire activity is substantially misrepresented. Lines 221 – 222 and 288-290 claim that in the majority of Alaska fire was projected to be near zero until the end of 2100... Fire in Alaskan tundra has been shown to be a rather common and extensive disturbance (see French et al 2015, Higuera et al 2011, He et al. 2019). And as the authors state, it is widely expected to increase by 2100 almost under any scenario (and definitely under the RCP8.5 considered in this study). It is definitely not occurring at the levels observed in the boreal forests but it is common particularly for the study area of this present project.

I don't believe that this will invalidate the major findings of this research that points to seed dispersal and fire occurrence as the dominant factors of shrubification compared to environmental suitability alone. However, it considerably increases the uncertainty of future projections considering that fire is poorly represented.

Authors: We thank the reviewer for the positive comment. We agree that multiple tundra fires were detected during the analyzed period of 1984-2014. However, as shown in the figure below, most fires were concentrated in areas close to the interface between the tundra (beige color) and the boreal forests (grey color). The averaged aerial fraction across the tundra is therefore low, especially in regions far away from the treeline. This is also the case in projected burn area (Extended Data Fig. 8b).

Fig. R2 | Areal fraction (0-1) of fire occurrence during 1984-2014 at a 4 km scale. Data from the annual product of differenced Normalized Burned Ratio (dNBR) at a 30 m resolution (Loboda et al. 2018). Grey areas are forests and other non-vegetated areas.

The low fire occurrence is consistent with previous results suggesting the fire return interval in Alaska is currently 200-600 years (Higuera et al. 2011; Yaping Chen et al. 2022). Even a projected increase of tundra fire by 300% increase (higher in some models) by 2100 still leads to a relatively small number of fires. Fire occurrence in tundra is unlikely to approximate what is currently happening in the boreal forest (Yang Chen et al. 2021). Nonetheless, to clarify the burn area is low but not zero, we clarified in lines 241-242: “In most of the tundra in Alaska

and northern Canada, the burn area was projected to be less than 3% until the end of the 21st century.”

Specific suggestions:

Lines 65-70: I believe this section is meant to indicate that in the Arctic seed dispersal is likely to occur over large distances. But I am not entirely sure. It might be helpful if the authors states explicitly their point here.

Authors: We added the summarizing sentence “In the Arctic, long-distance seed dispersal is a critical mechanism affecting species distribution.” for clarification (lines 65-66).

Line 72: Depending on definition of “historic”, this may not be entirely accurate. French et al 2015 show that over the past 80 years, large fires have been rather common in Alaska. Granted Alaska is substantially more impacted by fire than any other Arctic region, however, considering that this study focuses on Alaska and Westerns Canada, it should be acknowledged.

Authors: As described the above. While many fires occurred in the forest or in the tundra close to the treeline, the averaged areal fraction is relatively small (3.2%, lines 241-242). However, in lines 318-337, we highlighted the importance of correctly predicting fire and the interaction with shrubs given its large impact at occurrence.

Lines 120-123: If I understand the authors’ intent correctly, the Pearson’s r here is used to show how closely the model based entirely on environmental suitability can describe shrub distribution (essentially, how much variance it accounts for). However, the way the text reads it appears that the authors use this metric as a separate validation of their model’s performance which wouldn’t make a whole lot of sense considering that the model is derived based on the shrub distribution. Perhaps a slight clarification of the text would be beneficial.

Authors: Thanks for the suggestion. We rephrased the sentence to: “This pattern of suitability was largely consistent with observed shrub distribution in 1984 (Pearson’s $r = 0.92$, Fig. 1b), suggesting the environmental suitability alone explains shrub distribution under quasi-equilibrium conditions” (lines 129-132).

Fig 1: Please add the description units of suitability and the range to the figure caption.

Line 210: “... turning-off on factor...” should probably be “...turning off one factor...”

Authors: We corrected the typo to “turning off one factor” (line 228).

Citations:

- Bullock, James M., Steven M. White, Christel Prudhomme, Christine Tansey, Ramón Perea, and Danny A. P. Hooftman. 2012. "Modelling Spread of British Wind-Dispersed Plants under Future Wind Speeds in a Changing Climate." *The Journal of Ecology* 100 (1): 104–15.
- CAVM Team. 2003. *Circumpolar Arctic Vegetation (1:7,500,000 Scale), Conservation of Arctic Flora and Fauna (CAFF) Map No. 1*. Anchorage, Alaska: U.S. Fish and Wildlife Service.
- Chen, Yang, David M. Roms, Jacob T. Seeley, Sander Veraverbeke, William J. Riley, Zelalem A. Mekonnen, and James T. Randerson. 2021. "Future Increases in Arctic Lightning and Fire Risk for Permafrost Carbon." *Nature Climate Change*, April, 1–7.
- Chen, Yaping, Feng Sheng Hu, and Mark J. Lara. 2021. "Divergent Shrub-Cover Responses Driven by Climate, Wildfire, and Permafrost Interactions in Arctic Tundra Ecosystems." *Global Change Biology* 27 (3): 652–63.
- Chen, Yaping, Ryan Kelly, H el ene Genet, Mark Jason Lara, Melissa Lynn Chipman, A. David McGuire, and Feng Sheng Hu. 2022. "Resilience and Sensitivity of Ecosystem Carbon Stocks to Fire-Regime Change in Alaskan Tundra." *The Science of the Total Environment* 806 (Pt 4): 151482.
- Engler, Robin, Christophe F. Randin, Pascal Vittoz, Thomas Cz aka, Martin Beniston, Niklaus E. Zimmermann, and Antoine Guisan. 2009. "Predicting Future Distributions of Mountain Plants under Climate Change: Does Dispersal Capacity Matter?" *Ecography* 32 (1): 34–45.
- Finley, Andrew O., Sudipto Banerjee, and Alan E. Gelfand. 2015. "SpBayes for Large Univariate and Multivariate Point-Referenced Spatio-Temporal Data Models." *Journal of Statistical Software* 63 (February): 1–28.
- Fricke, Evan C., Alejandro Ordonez, Haldre S. Rogers, and Jens-Christian Svenning. 2022. "The Effects of Defaunation on Plants' Capacity to Track Climate Change." *Science* 375 (6577): 210–14.
- Hastie, Trevor, Jerome Friedman, and Robert Tibshirani. 2009. *The Elements of Statistical Learning: Data Mining, Inference, and Prediction*. Springer, New York, NY.
- Higuera, Philip E., Melissa L. Chipman, Jennifer L. Barnes, Michael A. Urban, and Feng Sheng Hu. 2011. "Variability of Tundra Fire Regimes in Arctic Alaska: Millennial-Scale Patterns and Ecological Implications." *Ecological Applications: A Publication of the Ecological Society of America* 21 (8): 3211–26.
- Ims, Rolf A., and John-Andr e Henden. 2012. "Collapse of an Arctic Bird Community Resulting from Ungulate-Induced Loss of Erect Shrubs." *Biological Conservation* 149 (1): 2–5.
- IPCC. 2018. *Global Warming of 1.5 C. An IPCC Special Report on the Impacts of Global Warming of 1.5 C above Pre-Industrial Levels and Related Global Greenhouse Gas Emission Pathways, in the Context of Strengthening the Global Response to the Threat of Climate Change, Sustainable Development, and Efforts to Eradicate Poverty*. Edited by Masson-Delmotte, V., P. Zhai, H.-O. P ortner, D. Roberts, J. Skea, P.R. Shukla, A. Pirani, W. Moufouma-Okia, C. P ean, R. Pidcock, S. Connors, J.B.R. Matthews, Y. Chen, X. Zhou, M.I. Gomis, E. Lonnoy, T. Maycock, M. Tignor, and T. Waterfield. Intergovernmental Panel on Climate Change.
- Lantz, Trevor C., Philip Marsh, and Steven V. Kokelj. 2013. "Recent Shrub Proliferation in the Mackenzie Delta Uplands and Microclimatic Implications." *Ecosystems* 16 (1): 47–59.

- Loboda, T. V., D. Chen, J. V. Hall, and J. He. 2018. "ABOVE: Landsat-Derived Burn Scar DNR across Alaska and Canada, 1985-2015."
- Mack, Michelle C., M. Sydonia Bret-Harte, Teresa N. Hollingsworth, Randi R. Jandt, Edward A. G. Schuur, Gaius R. Shaver, and David L. Verbyla. 2011. "Carbon Loss from an Unprecedented Arctic Tundra Wildfire." *Nature* 475 (7357): 489–92.
- Mielke, Konrad P., Tom Claassen, Michela Busana, Tom Heskes, Mark A. J. Huijbregts, Kees Koffijberg, and Aafke M. Schipper. 2020. "Disentangling Drivers of Spatial Autocorrelation in Species Distribution Models." *Ecography* 43 (12): 1741–51.
- Myers-Smith, Isla H., Bruce C. Forbes, Martin Wilking, Martin Hallinger, Trevor Lantz, Daan Blok, Ken D. Tape, et al. 2011. "Shrub Expansion in Tundra Ecosystems: Dynamics, Impacts and Research Priorities." *Environmental Research Letters: ERL [Web Site]* 6 (4): 045509.
- Myers-Smith, Isla H., Jeffrey T. Kerby, Gareth K. Phoenix, Jarle W. Bjerke, Howard E. Epstein, Jakob J. Assmann, Christian John, et al. 2020. "Complexity Revealed in the Greening of the Arctic." *Nature Climate Change* 10 (2): 106–17.
- Rogers, Haldre S., Noelle G. Beckman, Florian Hartig, Jeremy S. Johnson, Gesine Pufal, Katriona Shea, Damaris Zurell, et al. 2019. "The Total Dispersal Kernel: A Review and Future Directions." *AoB Plants* 11 (5): lz042.
- Schwalm, Christopher R., Spencer Glendon, and Philip B. Duffy. 2020. "RCP8.5 Tracks Cumulative CO₂ Emissions." *Proceedings of the National Academy of Sciences of the United States of America* 117 (33): 19656–57.
- Shipley, Benjamin R., Renee Bach, Younje Do, Heather Strathearn, Jenny L. McGuire, and Bistra Dilkina. 2022. "MegaSDM: Integrating Dispersal and Time-step Analyses into Species Distribution Models." *Ecography* 2022 (1). <https://doi.org/10.1111/ecog.05450>.
- Team, R. Core, and Others. 2013. "R: A Language and Environment for Statistical Computing." <http://r.meteo.uni.wroc.pl/web/packages/dplR/vignettes/intro-dplR.pdf>.
- Travis, Justin M. J., Maria Delgado, Greta Bocedi, Michel Baguette, Kamil Bartoń, Dries Bonte, Isabelle Boulangeat, et al. 2013. "Dispersal and Species' Responses to Climate Change." *Oikos* 122 (11): 1532–40.
- Wang, Jonathan A., Damien Sulla-Menashe, Curtis E. Woodcock, Oliver Sonnentag, Ralph F. Keeling, and Mark A. Friedl. 2020. "Extensive Land Cover Change across Arctic-Boreal Northwestern North America from Disturbance and Climate Forcing." *Global Change Biology* 26 (2): 807–22.
- Wilson, Scott D., and Christer Nilsson. 2009. "Arctic Alpine Vegetation Change over 20 Years." *Global Change Biology* 15 (7): 1676–84.

Reviewer comments, second round

Reviewer #2 (Remarks to the Author):

The authors have done a good and comprehensive set of responses and revisions in the light of my comments. I think the paper is much clearer now

Reviewer #3 (Remarks to the Author):

Thank you for undertaking a thorough review. I continue to be concerned that future fire extent is not adequately represented but this ultimately does not invalidate the findings of this work. And this paper presents important and insightful information which would benefit the community if it is published sooner rather than later.